# Fabrication of Polyelectrolyte Membranes of Pectin Graft-Copolymers with PVA and Their Composites with Phosphomolybdic Acid for Drug Delivery, Toxic Metal Ion Removal, and Fuel Cell Applications

**DOI:** 10.3390/membranes11100792

**Published:** 2021-10-18

**Authors:** Raagala Vijitha, Nagella Sivagangi Reddy, Kasula Nagaraja, Tiruchuru J. Sudha Vani, Marlia M. Hanafiah, Katta Venkateswarlu, Sivarama Krishna Lakkaboyana, Kummari S. V. Krishna Rao, Kummara Madhususdana Rao

**Affiliations:** 1Polymer Biomaterial Design and Synthesis Laboratory, Department of Chemistry, Yogi Vemana University, Kadapa 516005, Andhra Pradesh, India; vijitha.ragaala@gmail.com (R.V.); siva0406@gmail.com (N.S.R.); nagarajakasula33@gmail.com (K.N.); sudhatiruchuru@gmail.com (T.J.S.V.); 2Department of Earth Sciences and Environment, Faculty of Science and Technology, Universiti Kebangsaan Malaysia, Bangi 43600, Selangor, Malaysia; mhmarlia@ukm.edu.my; 3Centre for Tropical Climate Change System, Institute of Climate Change, Universiti, Kebangsaan Malaysia, Bangi 43600, Selangor, Malaysia; 4Laboratory for Synthetic & Natural Products Chemistry, Department of Chemistry, Yogi Vemana University, Kadapa 516005, Andhra Pradesh, India; kvenkat@yogivemanauniversity.ac.in; 5Department of Chemical Technology, Chulalongkorn University, Pathumwan, Bangkok 10330, Thailand; svurams@gmail.com; 6School of Chemical Engineering, Yeungnam University, 280 Daehak-Ro, Gyeongsan 38541, Gyeongbuk, Korea

**Keywords:** pectin, graft-copolymers, PEMs, drug delivery, adsorption, fuel fells

## Abstract

In this study, a simple method for the fabrication of highly diffusive, adsorptive and conductive eco-friendly polyelectrolyte membranes (PEMs) with sulfonate functionalized pectin and poly(vinyl alcohol)(PVA) was established. The graft-copolymers were synthesized by employing the use of potassium persulfate as a free radical initiator from pectin (PC), a carbohydrate polymer with 2-acrylamido-2-methyl-1-propanesulphonic acid (AMPS) and sodium 4-vinylbenzene sulphonate (SVBS). The PEMs were fabricated from the blends of pectin graft-copolymers (PC-g-AMPS and PC-g-SVBS) and PVA by using a solution casting method, followed by chemical crosslinking with glutaraldehyde. The composite PEMs were fabricated by mixing phosphomolybdic acid with the aforementioned blends. The PEMs were successfully characterized by FTIR, XRD, SEM, and EDAX studies. They were assessed for the controlled release of an anti-cancer drug (5-fluorouracil) and the removal of toxic metal ions (Cu^2+^) from aqueous media. Furthermore, the composite PEMs were evaluated for fuel cell application. The 5-fluorouracil release capacity of the PEMs was found to be 93% and 99.1% at 300 min in a phosphate buffer solution (pH = 7.4). The highest Cu^2+^ removal was observed at 206.7 and 190.1 mg/g. The phosphomolybdic acid-embedded PEMs showed superior methanol permeability, i.e., 6.83 × 10^−5^, and 5.94 × 10^−5,^ compared to the pristine PEMs. Furthermore, the same trend was observed for the proton conductivities, i.e., 13.77 × 10^−3^, and 18.6 × 10^−3^ S/cm at 30 °C.

## 1. Introduction

Polymeric membranes have attracted much attention because of their unique advantages in the fields of biotechnology, food industry, water treatment and fuel cells [1,2,3,4,5]. There are various kinds of polymer membranes available, i.e., pure polymer, blend, mixed matrix, polyelectrolyte, composite, organic-inorganic hybrid, etc., and which are nonionic, ionic, or complex in nature, to meet day-to-day challenges [6,7,8]. Recently, castor oil-based polyurethane and polyaniline membranes have been used for the application of separation technology [8]. Alginate-based silver nano composite hydrogel membranes have been developed for colon-specific drug delivery of 5-fluorouracil (5-FU) [9]. Furthermore, a variety of MXene/polymer membranes have been developed for potential industrial applications [10]. The production of polymer membranes from renewable sources is currently of interest due to the fact that this natural polymer reduces/replaces the need for synthetic polymers. However, polyelectrolyte membranes (PEMs) have great advantages over other membranes due to their tunable characteristics, such as environmental responsivity, ion selection, solvent resistance, and ease of fabrication. To meet this criterion, various polysaccharide-based membranes have been extensively studied for drug delivery [11], tissue engineering [12], pervaporation [13], desalination [14], COD removal [15], bioenergy generation [15], toxic metal ion removal [16], fuel cell [14,15,17].

Pectin (PC) is a natural polysaccharide comprised of repeated units of α-(1,4)-linked-d-galacturonic acid with varying degrees of methylated carboxylic acid [9]. PC is frequently used for the development of membranes for various applications since it enhances hydrophilicity. Its applications are benign due to its environmental benefits, such as its biodegradability and natural availability [18,19,20,21,22]. However, PC has some disadvantages, including uncontrolled hydrostability, low functionality, and its non-flexible membrane formation. To overcome these problems, pectin has been chemically grafted with 2-acrylamido-2-methyl-1-propanesulphonic acid (AMPS) and sodium 4-vinylbenzene sulphonate (SVBS), resulting in graft-copolymers that are blended with polyvinyl alcohol (PVA). PVA is widely used in membrane preparation due to its extensive properties, such as its good film formation, high hydrophilicity, and high chemical resentence. PVA-based membranes, in particular, are used for ethanol dehydration as ethanol-water azeotropes due to the selective passage of water molecules over methanol and ethanol [23,24,25].

PEM-based drug delivery systems have recently gained in popularity, particularly in the biomedical fields of transdermal drug delivery, wound healing, and tissue engineering [26]. Industries such as organic dyes, metal cleaning, mining, and metal finishing emit toxic heavy metal pollutants into the environment [27]. When humans consume these toxins, they can have serious consequences, such as brain damage and cancer. Hence, removing toxic metal ions and volatile organic compounds from wastewater in a sustainable manner is essential [27,28]. The ion exchange process, the deposition of metals at electrodes, precipitation (coagulation), electrocoagulation, and filtration through membranes are all effective methods for removing toxic heavy metals from the environment [29,30,31,32,33,34,35]. However, these technologies take longer to remove harmful metals and other waste products from wastewater and are more expensive [33,34,35]. Fuel cells are clean energy sources based on their high density and low emissions performance for various applications, such as electric vehicles, portable electronics, and residential electricity generation [36,37,38]. As far as their applications are concerned, they are intended as energy sources to extend stationary applications. Several promising proton-conductive polyelectrolyte membranes have been established for proton-exchange membrane fuel cells and direct methanol fuel cells [39]. An efficient fuel cell membrane must allow protons to move quickly; hence, researchers moved their focus towards the development of potential cation exchange membranes [40,41], which led to the conversion of chemical energy into electrical energy with high efficiency and low emissions of pollutants. Consequently, fuel cells have received considerable attention.

Previously, we successfully developed chitosan-, sodium alginate-, pectin-, and PVA-based matrices for pervaporation, drug delivery, and toxic metal ion removal. In a continuation of our research, the present work demonstrates the synthesis of pectin graft-copolymers (PC-g-AMPS and PC-g-SVBS) by simple free radical polymerization and the fabrication of new PEMs from PVA with pectin graft-copolymers by using a solution costing method; in addition, composite PEMs were fabricated by using PVA-pectin graft-copolymers embedded with polyacid (PMA). To the best of tour knowledge, there are currently no studies on PVA/pectin graft-copolymers for drug delivery and Cu^2+^ ion removal, nor on PMA-embedded PEMs from PVA-pectin graft-copolymers for fuel cell applications.

## 2. Experimental

### 2.1. Materials

The following were obtained from Aldrich chemicals, India: 2-Acylamido-2-methylpropane sulfonic acid (AMPS), sodium 4-vinylbenzenesulfonate (SVBS), poly(vinyl alcohol), (PVA, MWt.: 75,000), potassium persulfate (KPS), and phosphomolybdic acid (PMA), 5-fluorouracil (5-FU). Pectin (MWt.: 30,000–100,000), glutaraldehyde, hydrochloric acid, methanol and acetone were obtained from Merck Chemicals, India. Cupric nitrate trihydrate, extra-pure grade, was obtained from FINAR chemicals, India. Graft-copolymers (PC-g-AMPS and PC-g-SVBS) were synthesized as per our laboratory’s previous work [42]. All the experiments were carried out by using double-distilled water.

### 2.2. Fabrication of PEMs

We placed required amount of PVA (3 g) in a 250 mL of a beaker, added 80 mL of double-distilled water, and stirred for 4 h with heating at 80 °C to obtain a clear PVA solution. To this, a 20 mL solution of graft-copolymers (2 g of PC-g-AMPS/PC-g-SVBS) was added and stirred overnight to obtain a clear solution. The blended solutions were poured onto a clean glass plate in a dust-free environment and allowed to dry for about 24 h. The dried membranes peeled from the glass plate and they were immersed in a cross-linking mixture (double distilled water:acetone (50:50) + glutaraldehyde (2.5 mL) + HCl (1 mL)) for about 4 h. A cross-linking reaction occurred between the -OH of the polymer chains and the -CHO of the glutaraldehyde (GA). Finally, the membranes were washed multiple times with double-distilled water to remove the residual GA and solvent. These membranes are named PPCAM and PPCSB. Furthermore, the membranes embedded with PMA (10% *w*/*w*) were prepared using the same procedure described above and named PPCAM-PMA and PPCSB-PMA. Here, a pure PVA membrane was prepared without adding graft copolymer or PMA (Figure 1).

### 2.3. Swelling Studies

All of the membranes were subjected to water-uptake measurements. The dry membrane weights (*W_d_*) were measured and equilibrated in double-distilled water for 24 h at room temperature. Surface blotting was performed on the membranes carefully with tissue paper, and the sorbed membranes were weighed (*W_s_*). The following expression was used to calculate the percentage of equilibrium swelling ratio (%*ESR*) value for the aforementioned membranes.
(1)$$\%ESR=\frac{W_{s}-W_{d}}{W_{d}}\times 100$$

### 2.4. 5-Fluorouracil Encapsulation Efficiency and In Vitro Drug Release Studies

The physical absorption method was used to load 5-FU into the PPCAM-, PPCSB-, and PVA-dried membranes. The 5-FU solution was prepared by mixing 5-FU and equimolar sodium hydroxide in double-distilled water [20]. A total of 200 mg of PEMs were placed into 10 mL of 5-FU solution and allowed to swell for 12 h at room temperature for physical adsorption of the drug. To evaluate the encapsulation efficiency, 20 mg of drug-loaded dried PEMs were placed in 10 mL of phosphate buffer solution (PBS) for 12 h, and the resulting membrane was crushed in an agate mortar. The filtrate was subjected to UV-Vis spectrophotometer (LABINDIA, UV-3092) analysis to calculate the %5-FU loading and encapsulation efficiency (EE) using the equations below:(2)$$\left(\%\right)\;Drug\;loading=\left(\;\frac{Amount\;of\;5-FU\;loaded\;in\;to\;PEMs}{Amount\;of\;PEMs\;taken}\right)\times 100\;$$
(3)$$\left(\%\right)Encapsulation\;efficiency\;\;=\left(\;\frac{Actual\;loading\;of\;5-FU}{Theoretical\;loading\;of\;5-FU}\right)\times 100$$

The 5-FU drug released from the PPCAM, PPCSB, and PVA was tested using a dissolution tester apparatus (LAB INDIA, DS-8000, Mumbai, India) at a constant rotation speed of 100 rpm under two different pH conditions, pH-1.2 and pH-7.4 (simulated gastric and intestinal conditions). The samples were analyzed with a UV-Vis spectrophotometer (LABINDIA, UV-3092) at a fixed wavelength of 5-FU, 270 nm. The sample measurements were taken three times to calculate the standard deviation.

The kinetic models, such as the Korsmeyer–Peppas, zero order, first order, Higuchi, and Hixson–Crowell models [43], were used to analyze the in vitro drug release data as per the equations given below:(4)$$\frac{M_{t}}{M_{\infty }}=Kt^{n}$$
(5)$$Q=Q_{0}-K_{0}t$$
(6)$$lnQ=lnQ_{0}-K_{1}t$$
(7)Mt=KH12t
(8)Q13=Q013KCt

Here, *M_t_* and *M_∞_* are the fractions of the drug concentration released from the PEMs at a time *t* and ∞, respectively; *k* is the 5-FU release rate constant, and *n* is the 5-FU drug release exponent; *M_t_* is the amount of 5FU release at time *t*, *K_H_* is the Huguchi rate constant, *Q* is the amount of the 5FU release at time *t*, *Q*_0_, is the amount of 5FU release time at *t*, *K*_1_ is the rate constant of first order, *K_0_* is the rate constant of zero order, and *K_C_* is the 5FU release constant of Hixson-Crowell cube.

### 2.5. Copper Ion Removal

The batch adsorption method was used for the metal ion sorption experiments [16,21]. Small pieces of PEMs were weighed and placed in 20 mL of various concentrations of Cu^2+^ ion solution. The membranes were removed from the solution after 24 h, and the remaining solution was subjected to atomic absorption spectroscopy (SHIMADZU, AA-6880) analysis to determine the equilibrium concentrations of the metal ions after sorption. The following equation was used to calculate the equilibrium sorption amount (*Q_e_*) of the PEMs:(9)$$Qe=\frac{(C_{o}-C_{e})\;V}{M}$$
where *C_o_* is the initial metal ion concentration, *C_e_* is the metal ion equilibrium concentration, *V* is the volume of metal solution used in the sorption, and *M* is the weight of the dried membrane.

### 2.6. Ion Exchange Capacity, Proton Conductivity, and Methanol Permeability Studies

The ion-exchange capacity (IEC) of the various membranes used in this study was determined using simple acid-base titration [44]. The membrane was immersed in 50 mL of 3 M NaCl solution for 24 h. To convert the H^+^ ion of the membranes into Na^+^, the resultant solution was titrated against 0.01 N NaOH using the phenolphthalein indicator. The value of IEC (meq/g), was calculated by the following equation:(10)$$\mathrm{IEC}=\frac{Volume\;of\;NaOH\;consumed\times Normality\;of\;NaOH}{W_{d}}$$

Proton conductivity measurements were taken for various PEMs fabricated as per the research [45] using the four-probe impendence technique (BekkTech conductivity cell). The membrane sample was clamped between four platinized probes, which were held together by a Teflon block. The cell was placed in a humidified chamber (98%). The cell was then connected to an impedance analyzer (biologic, SP-150). The following equation was used to calculate the ionic conductivity data for the membranes:(11)$$\sigma =\frac{L}{RA}$$
where *L* is the thickness of membrane, *R* is the measured resistance, and *A* is the area of the membrane.

A simple side-by-side diffusion cell was used to test the methanol permeability of all the membranes. The membranes were equilibrated in water for 24 h before being clamped between the well-stirred donor (A) and receptor (B) compartments, with a 3.19 cm^2^ membrane cross-sectional area exposed to the solutions in both compartments. The donor compartment (V_A_ = 80 mL) was initially charged with a methanol solution, while the receptor compartment (V_B_ = 80 mL) was initially filled with water. The diffusion cell was kept in a thermostat at 30 °C. The methanol flux across the membrane was caused by the concentration difference between the two compartments. The methanol concentration in the receptor compartment was calculated using a refractive index (RI) measurement taken at regular intervals with a refractometer AR4 (A. Kruss Optronic, Hamburg, Germany). The methanol concentration was determined using a graphical interpretation of the slope made from methanol concentration against time [46]:(12)$$\mathrm{Permeability}P=\frac{C_{b}V_{b}X}{C_{a}A\;T}$$
where *C_b_* is the concentration of methanol in the receptor compartment, *V_b_* is the receptor volume, *X* is the thickness of the membrane, *C_a_* is the concentration of methanol in the donor compartment, *A* is the area of the membrane, and *T* is the time taken to equilibrate the system.

### 2.7. Characterization

We performed ^1^H nuclear magnetic resonance (NMR, Brucker A Vance 500 MHz, Billerica, USA) studies of pectin graft-copolymers in D_2_O with tetramethylsilane as an internal standard. Fourier transform infrared studies were performed using a PerkinElmer (Spectrum Two Model, Singapore). The sample pellets were prepared with KBr under a pressure of 600 dyn/m and recorded in the region of 4000–400 cm^−1^ with a scan rate of 8 per sample. Furthermore, attenuated total reflection-FTIR (Bruker, Alpha-II, Eco-ATR, Billerica, MA, USA) was used to characterize the drug-loaded and copper metal ion-adsorbed PEMs. The differential scanning calorimetry of the graft-copolymers was performed with TA instruments (STA, Q600, New Castle, DE, USA) in a nitrogen atmosphere. The X-ray diffraction studies were performed using Rigaku (miniflex 600, Tokyo, Japan) with a scanning speed of 0.01–100°/min in the range of 2θ value 10–80°. The surface morphology images of the PEMs were taken by JOEL scanning electron microscopy (SEM, JSM IT500, Tokyo, Japan). Furthermore, energy-dispersive X-ray spectroscopy was used to characterize the drug-loaded and copper metal ion-adsorbed PEMs before scanning with SEM, and the membranes were coated with gold using sputtering equipment.

## 3. Results

### 3.1. Synthesis of Graft-Copolymers

Pectin copolymers with AMPS and SVBS were synthesized by simple free radical polymerization using potassium persulphate as the initiator. Pectin-g-AMPS and pectin-g-SVBS are characterized by FTIR, ^1^H NMR, XRD, and DSC (Appendix A). Optimized reaction parameters, such as the temperature, reaction time, initiator concentration, and monomer concentration of PC-g-AMPS PC-g-SBVS are shown in Appendix A and Appendix A. The optimized conditions for PC-g-AMPS are reaction temperature 60 °C, AMPS concentration 3 mmol, KPS concentration 0.07 mol., and reaction time 150 min. The highest %grafting and %grafting efficiency observed were 82.16 and 66.07, respectively. In pectin-g-SVBS, the optimized reaction conditions for PC-g-SBVS were: reaction temperature 65 °C, SVBS concentration 3 mmol, KPS concentration 0.07 mol., reaction time 120 min. The highest %grafting and %grafting efficiency observed were 67.03 and 53.90, respectively.

### 3.2. FTIR Studies

Figure 2 shows the FTIR spectra of PPCAM, PPCAM-PMA, PPCSB, and PPCSB-PMA. The well-known significant peak was observed at 3200 to 3500 cm^−1^ of -OH stretching vibrations of PVA and pectin; furthermore, a significant peak was observed for the acetal ring, C-O-C, at 1097 cm^−1^, which was responsible for the crosslinking reaction between the -CHO groups of the glutaraldehyde and the -OH functional groups of PVA and graft-copolymers (for all PEMs). The stretching vibrations of pectin were observed at 1735 cm^−1^. The symmetric stretching vibrations of the -SO_3_H groups of AMPS and SVBS were observed at 1087 and 1022 cm^−1^, respectively. The stretching vibrations of C=C of the SVBS benzene ring were observed at 1582 and 1495 cm^−1^. In the case of the PMA-embedded composite PEMs, in addition to the above peaks, additional peaks were observed at 783, 821, 964, and 1060 cm^−1^for the Mo-O-Mo (edge sh), Mo-O-Mo (corner sh), Mo=O (terminal), and P-O groups, respectively, for the PMA. Furthermore, the attenuated total reflection-FTIR used to characterize the 5-FU-loaded and copper metal ion-adsorbed PEMs. However, significant peaks of 5-FU were observed at 1265, 821, and 792 cm^−1^ for the bonds of C-F and C-H (Figure 2(iC,iiC)). In the case of the Cu^2+^ ion-adsorbed PEMs, the characteristic peaks of -NH, -COO- and -SO_3_^-^ shifted to high wavelengths (Figure 2(iE,iF,iiE,iiF)). These results indicate the presence of 5-FU and Cu^2+^ ions in the PEMs.

### 3.3. XRD Studies

To understand the crystalline behavior of all four PEMs (PPCAM, PPCAM-PMA, PPCSB, and PPCSB-PMA), they were characterized using XRD analysis. As can be seen from Figure 3, the pristine membranes of the PVA-blend pectin graft-copolymers (PPCAM and PPCSB) presented a semi-crystalline structure with a large peak at the scattering angle [47], which was approximately 2θ value at 20°. The presence of PMA contributed significantly to the crystallinity of the PPCAM-PMA and PPCAM-PMA composite PEMs. The PMA in the nanocomposite PEMs caused the broadening of the XRD peaks. This property indicated that the crystallinity of the composite PEMs decreased as the PMA content was present in the membranes. The decrease in crystallinity of the composite PEMs had a significant impact on the conductivity of the composite membranes.

### 3.4. SEM and EDAX Studies

Figure 4 shows the SEM images of pristine (PPCAM & PPCSB), PMA-embedded (PPCAM-PMA & PPCSB-PMA), 5-FU-encapsulated (PPCAM & PPCSB), and Cu^2+^ ion-adsorbed (PPCAM & PPCSB) PEMs. In comparison to the pristine membranes (Figure 4A,B), the morphology of the PMA-embedded PEMs (Figure 4C,D) was rough, and the PMA particulates were finely dispersed. In the case of the 5FU-encapsulated PEMs (Figure 4E,F), considerable needle-like crystal formations of 5-FU were detected on the surface of PPCAM and PPCSB, which may have been related to the sorption and desorption phenomena of the swollen- and dry-state PEMs, respectively. The PEMs of Cu^2+^ ion-adsorbed PPCAM and PPCSB displayed similar results (Figure 4G,H); however, their surfaces were densely packed with dendritic copper atom micro/nanostructures. The EDAX spectra in Figure 5 confirm the presence of 5-FU in the PPCAM and PPCSB PEMs. According to the EDAX patterns (Figure 5), the surface of the membranes contained the elements C, N, O, F, Na, and S; the weight percentages of fluorine were 7.15 and 4.99, depending on the intensity of the fluorine peak. Furthermore, the elements present in the Cu^2+^ ions adsorbed PPCAM, and the PPCSB PEMs were C, N, O, S, and Cu; the weight percentages of copper were 25.31 and 18.66, respectively, depending on the intensity of the copper peak.

### 3.5. Water Uptake Measurements

The equilibrium water uptake capacity of the PEMs was measured using a gravimetric method in double-distilled water at 30 ± 0.5 °C, and the results are shown in Table 1. According to the data, the maximum water uptake observed was 30 min for three membranes, and for the PPCSB-PMA membrane, it was around 60 min. Table 1 shows the equilibrium water uptake capacity of all PEMs. The water uptake values ranged from 72 to 340. It can be seen that the equilibrium water uptake was greatest for both pristine membranes (PPCAM & PPCSB). However, the PMA-embedded PEMs exhibited low water uptake, which could be attributed to strong chemical interactions and intermolecular bonds between the polymer and the inorganic filler [48,49].

### 3.6. 5-Fluorouracil Release Studies 

The in vitro drug release studies of 5-FU-loaded PPCAM and PPCSB membranes were described in terms of pH condition in simulated pH conditions, i.e., pH-1.2 and pH-7.4 (gastric & intestinal) at 37 °C, as shown Figure 6. The 5-FU release profiles indicated that drug release from the PPCAM and PPCSB membranes was slightly higher in the pH-7.4 region when compared to the pH-1.2 region. However, in the case of the PVA, the pH was not significantly influenced by the 5-FU release profiles. The 5-FU-loaded PVA cumulative drug release at both mediums was found to be 91.50% at 30 min, 94.08% at 60 min, 94.84% at 120 min, and 96.58% at 180 min. The PPCAM cumulative drug release (pH-1.2) was observed as 8.54% at 30 min, 27% at 60 min, 47.41% at 180 min, and 98.5% at 300 min. The PPCSB membrane where the cumulative drug release medium simulated the gastric fluid region (pH-1.2) was found to be 6.8% at 30 min, 19.68% at 60 min, 60.47% at 180 min, and 93% at 300 min. The 5-FU release at pH -7.4 from the PVA membrane indicated that 91.5% was released at 30 min, 94.6% at 60 min, 96.5% at 180 min, and 99.1% at 300 min. The 5-FU release observed from the PPCAM membrane was found to be 21.6% at 30 min, 53.9% at 60 min, 88.6% at 180 min, and 99.2% at 300 min. The 5-FU release observed from the PPCSB membrane was found to be 12.51% at 30 min, 45% at 60 min, 77.7% at 180 min and 93% at 300 min.

The in vitro drug release profiles of PVA, PPCAM, and PPCSB were evaluated by using kinetic models, such as Korsmeyer–Peppas, zero order, first order, Higuchi, and Hixson–Crowell. Among all these, the release profiles were best fitted with the Korsmeyer–Peppas and the Hixson–Crowell kinetic models [20]. The *k* and *n* values were calculated, and the first case, *n* < 0.5, showed that the polymer network followed fickian diffusion. In the second case, values between 0.5 < *n* < 0.89 corresponded to non-fickian diffusion and an anomalous mechanism. The third case, *n* = 1, indicated completely non-fickian and super case II drug release kinetics. In this study, the exponent values (*n*) ranged between 0.40 and 0.81, followed by the completely non-fickian diffusion transport mechanism. The PPCAM and PPCSB drug release kinetics models displayed good fits in the Korsmeyer–Peppas and Hixson–Crowell models.

The commercial low-methaxyl pectin and de-esterified pectin films exhibited two steps of clindamycin release. Both compositions released 100% clindamycin within 60 min [50]. The pectin-montmorillonite hybrid films exhibited a higher cumulative drug release rate (70%) in the first half-hour, with a poor sustained release. The 5-FU release rates were well fitted with the Elovich model compared to other models. The Ritger–Peppas model and the first-order model were applied to drug release to describe the drug release process in the alkaline region [51]. This resulted in multiple drug release mechanisms: (1) the membrane surface of 5-FU released in the simulated environment, (2) the membrane layer breaking of the PEM induced drug release in the 5-FU, (3) the high concentration of the 5-FU in the inner layer drove drug release, (4) the polymer–solvent interactions, as well as the polymer–solute interactions, meant that the pectin-Ca^+2^-based films displayed a higher drug release rate, i.e., 63.89%, in simulated intestinal fluid [52].

### 3.7. Copper Metal Ion Adsorption Studies

The Cu^2+^ ion adsorption studies of the PPCAM and PPCSB membranes were performed at optimized conditions, with a time of 120 min and a pH of 5.5. Figure 7 depicts the relationship between the initial metal ion concentration and the amount of PEM. The fabricated membranes were used to perform Cu^2+^ adsorption at various initial metal ion concentrations (400–1100 ppm) while keeping the optimum time and pH. The adsorption capacity of the PPCAM and PPCSB membranes increased with increasing initial metal ion concentration and then plateaued at higher concentrations due to the saturation membrane chelating sites. The metal ion adsorption phenomenon observed was that the Cu^2+^ uptake increased as the initial concentration of metal ions increased. This was due to the overwhelming mass transfer resistance of the ions between the adsorbent and bulk fluid phases. Furthermore, increasing the concentration of metal ions increased the number of collisions between the metal ions and the adsorbent, which improved the adsorption process [53].

The high adsorption efficiency may have been attributable to the presence of sulfite (SO_3_^2−^), hydroxyl (-OH), amine (-NH), and carboxyl (-COO^-^) groups present in the PEMs. These functional groups have a high affinity towards metal ions and act as binding sites, indicating that they are responsible for Cu^2+^ adsorption. The maximum adsorption capacities of the membranes PPCAM and PPCSB for Cu^2+^ were determined to be 206 and 190 mg/g, respectively. It is possible to predict that metal ion uptake will be based on adsorption and complexation (chelation). As a result, PPCAM and PPCSB membranes can be used to easily separate Cu^2+^ ions.

Table 2 compares the adsorption capacity (Cu^2+^ ions) of the current work to that observed in other studies. According to these, pectin-based films have an adsorption capacity of 29.2 mg/g [53], pectin-iron oxide magnetic nanocomposites have an adsorption capacity of 48.99 mg/g [54], and pectin/poly(AM-co-AGA) hydrogels have an adsorption capacity of 203.7 mg/g [21]. The present work demonstrates that the adsorption capacity values of Cu^2+^ ions reported are significantly greater than those reported in previous research: 206 and 190 mg/g. This observation suggests that the process of functionalizing PEMs facilitates the significant adsorption of bivalent metal ions.

### 3.8. Proton Conductivity and Methanol Permeability Studies

Figure 8A depicts the proton conductivity of PEMs. The proton conductivity of the pristine PEMs was in the order of 10^−3^ S/cm. The PMA-embedded PEMs exhibited a significant increase in proton conductivity. This was due to the presence of inorganic filler phosphomolybdic acid and functional groups, such as -COOH of pectin and -NH & -SO_3_H of AMPS or SVBS, in the PEMs; these groups aided in the development of easy hydrogen bonding to the polymer network, which led to proton conductivity via the PEMs. Furthermore, this was supported by the PEMs’ ion exchange capacity (IEC) and percentage of equilibrium swelling ratio (%ESR), with the proton conductivity being dependent on the IEC and %ESR. The proton conductivity of all the PEMs was shown to be between 13.77 × 10^−3^ and 18.6 × 10^−3^ S/cm at 30 °C. The proton conductivity of the PEMs increased as the water content in the polymer network rose; this may have bene due to the creation of water channels or domains, which enhance proton transport through the PEMs.

Figure 8B depicts the methanol permeability of the PEMs. When compared to the pristine PEMs, the methanol permeability of the PMA-embedded PEMs was low. The PMA-embedded PEMs displayed superior methanol permeability, i.e., 6.83 × 10^−5^ and 5.94 × 10^−5^, compared to the pristine PEMs. This could have been due to PMA’s pseudo liquid phase characteristic [55], which allows methanol to be easily absorbed by the solid matrix and water molecules in the secondary structure of the PMA to be replaced by methanol. As a result, methanol molecules are not allowed to flow through the membrane [56].

## 4. Conclusions

In this study, we fabricated polyelectrolyte membranes of pectin graft-copolymer with PVA by using a simple solution casting method. These membranes exhibited good swelling, diffusion, and adsorption characteristics. PEMs are used for anti-cancer drug delivery and Cu^2+^ ion removal; furthermore, PMA-embedded PEMs are used for fuel cell applications. The 5-FU release characteristics of the PEM demonstrate that the release kinetics followed the non-Fickian mechanism and the release of 5-FU was influenced by the pH. Based on the metal ion removal studies, the maximum adsorption capacity of the Cu^2+^ ion was 206 ± 6.1 and 190 ± 7.8 mg/g with PPCAM and PPCSB PEMs, respectively. Phosphomolybdic acid-embedded PEMs displayed significant fuel cell characteristic, such as high proton conductivity (17.01 ± 0.71 × 10^−3^ and 18.6 ± 0.74 × 10^−3^ S/cm) and low methanol permeability (2.77 ± 0.14 × 10^−5^ and 5.94 ± 0.19 × 10^−5^ cm^−2^/s). Overall, the present study demonstrates that PEMs are superior to pure polymers as matrices for the controlled release of anti-cancer drugs, the separation of bivalent metal ions from aqueous media, proton conductivity, and methanol permeability. Finally, the results suggest that the fabrication of PEMs with low cost and easy functionalization using green building materials could pave the way towards advanced material technologies for the benefit of society and industry.

## Figures and Tables

**Figure 1 membranes-11-00792-f001:**
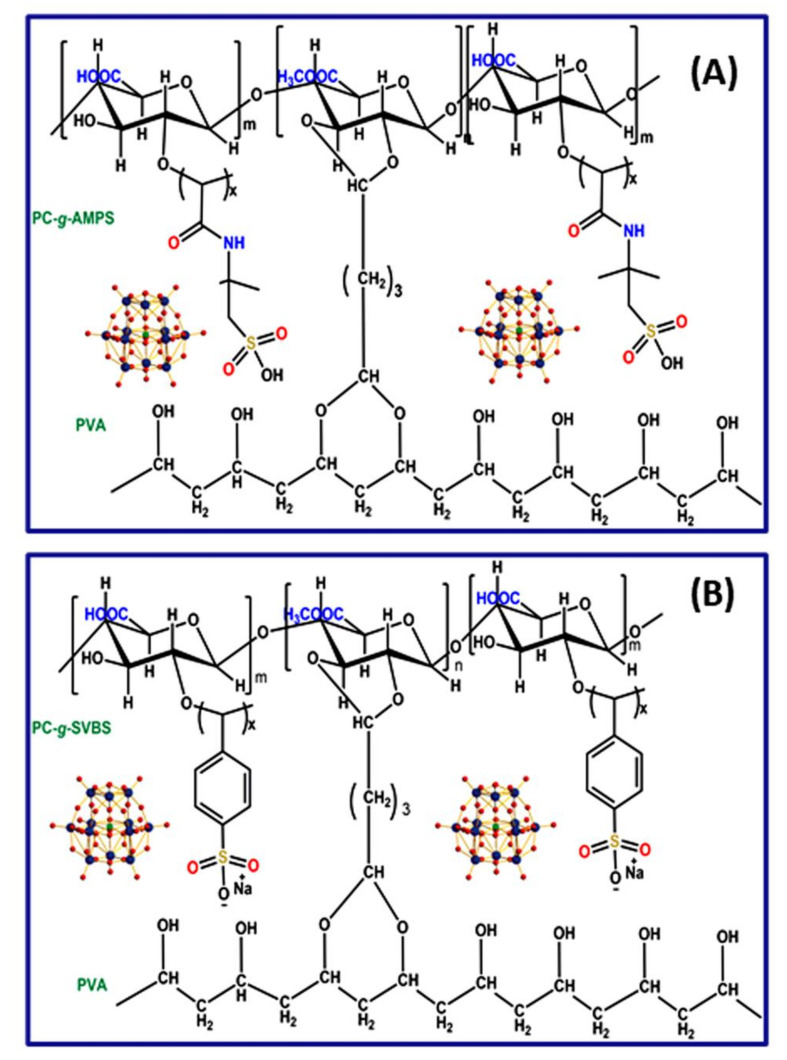
Schematic representation of PMA-embedded PEMs: PPCAM (**A**) and PPCSB (**B**).

**Figure 2 membranes-11-00792-f002:**
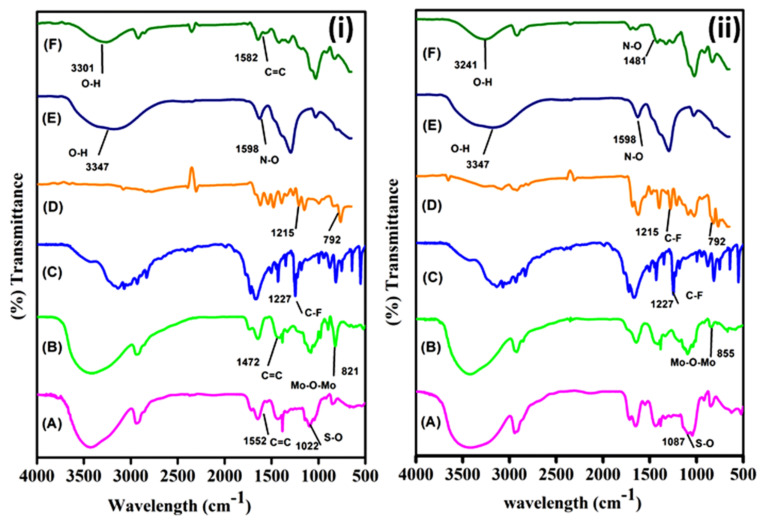
FTIR spectra of (**i**) PPCAM (A), PPCAM-PMA (B), pure 5-FU (C), PPCAM-5-FU (D), cupric nitrate trihydrate (E), and PPCAM-Cu^2+^ (F); and (**ii**) PPCSB (A), PPCSB-PMA (B), pure 5-FU (C), PPCSB-5-FU (D), Cupric nitrate trihydrate (E), and PPCSB-Cu^2+^ (F).

**Figure 3 membranes-11-00792-f003:**
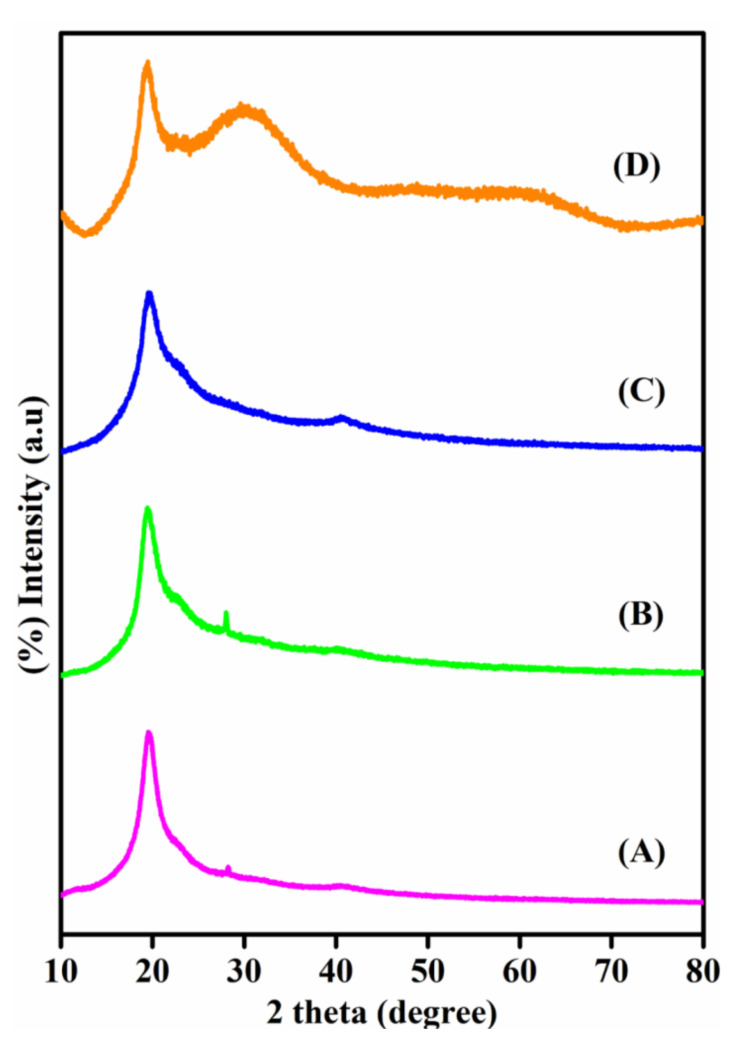
XRD patterns of the PPCAM (A), PPCSB (B), PPCAM-PMA (C), PPCSB-PMA (D) membranes.

**Figure 4 membranes-11-00792-f004:**
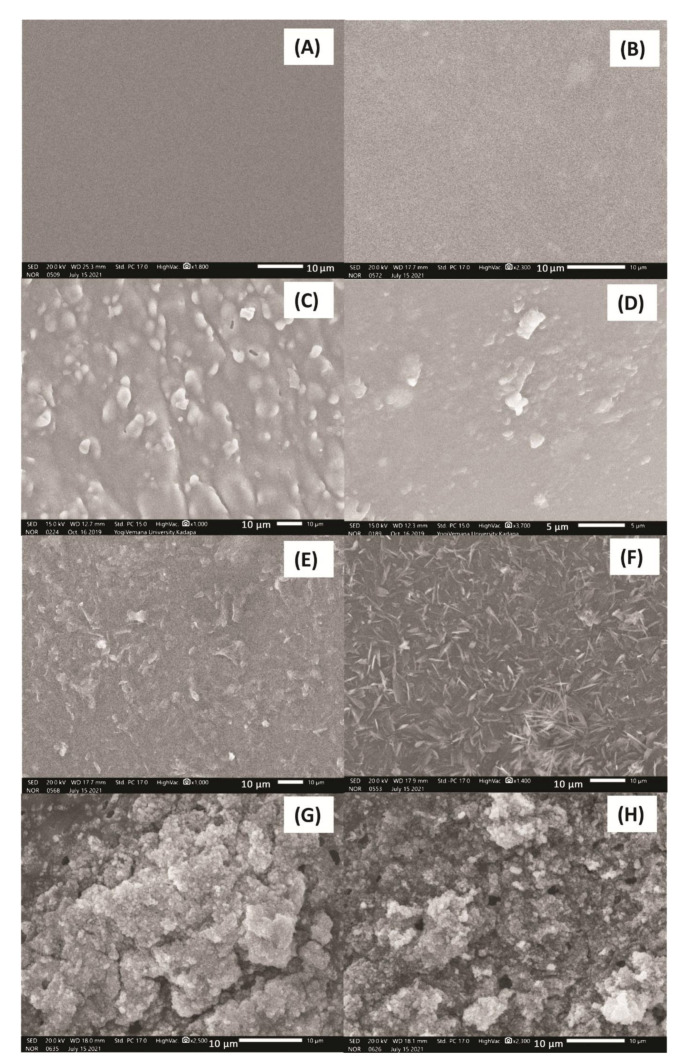
SEM images of pristine PPCAM (**A**), PPCSB (**B**), PPCAM-PMA (**C**), PPCSB-PMA (**D**), 5-FU-encapsulated PPCAM (**E**), 5-FU-encapsulated PPCSB (**F**), Cu^2+^-adsorbed PPCAM (**G**), and Cu^2+^-adsorbed PPCAM (**H**).

**Figure 5 membranes-11-00792-f005:**
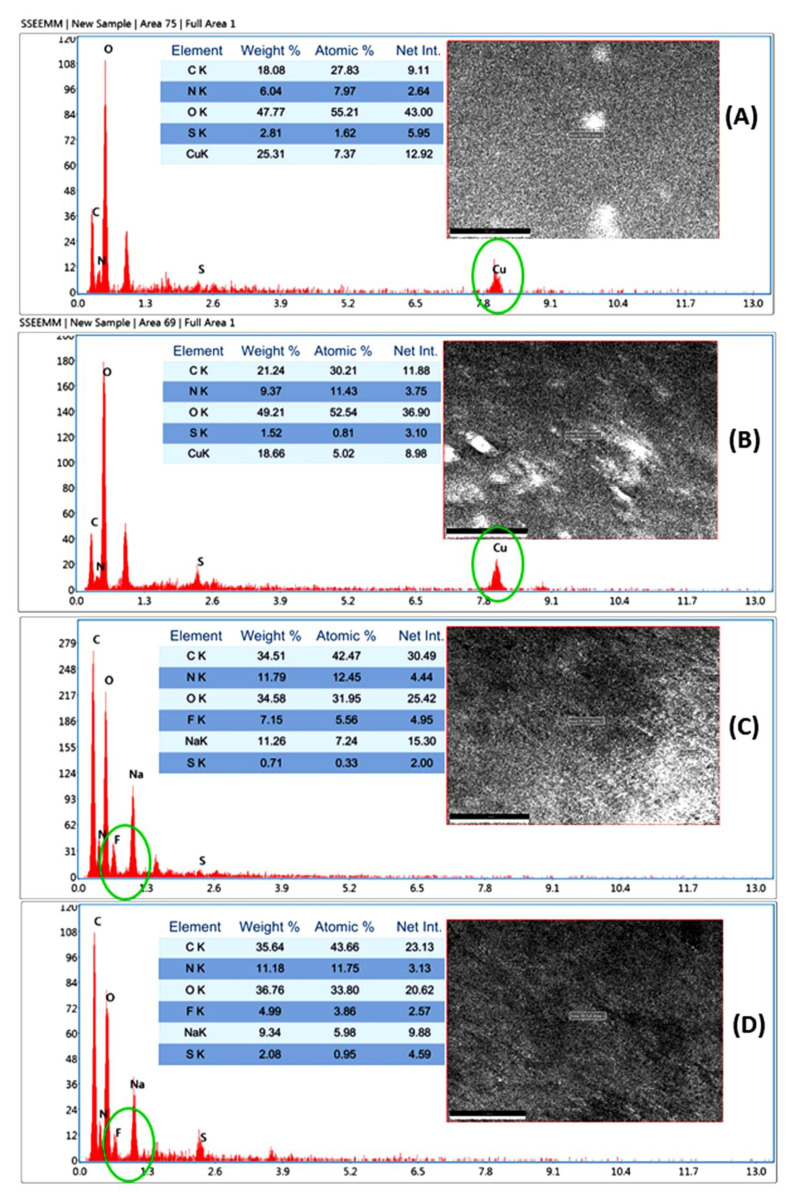
EDAX images of Cu^2+^-adsorbed PPCAM (**A**), and Cu^2+^-adsorbed PPCSB (**B**), 5-FU-encapsulated PPCAM (**C**), and 5-FU-encapsulated PPCSB (**D**).

**Figure 6 membranes-11-00792-f006:**
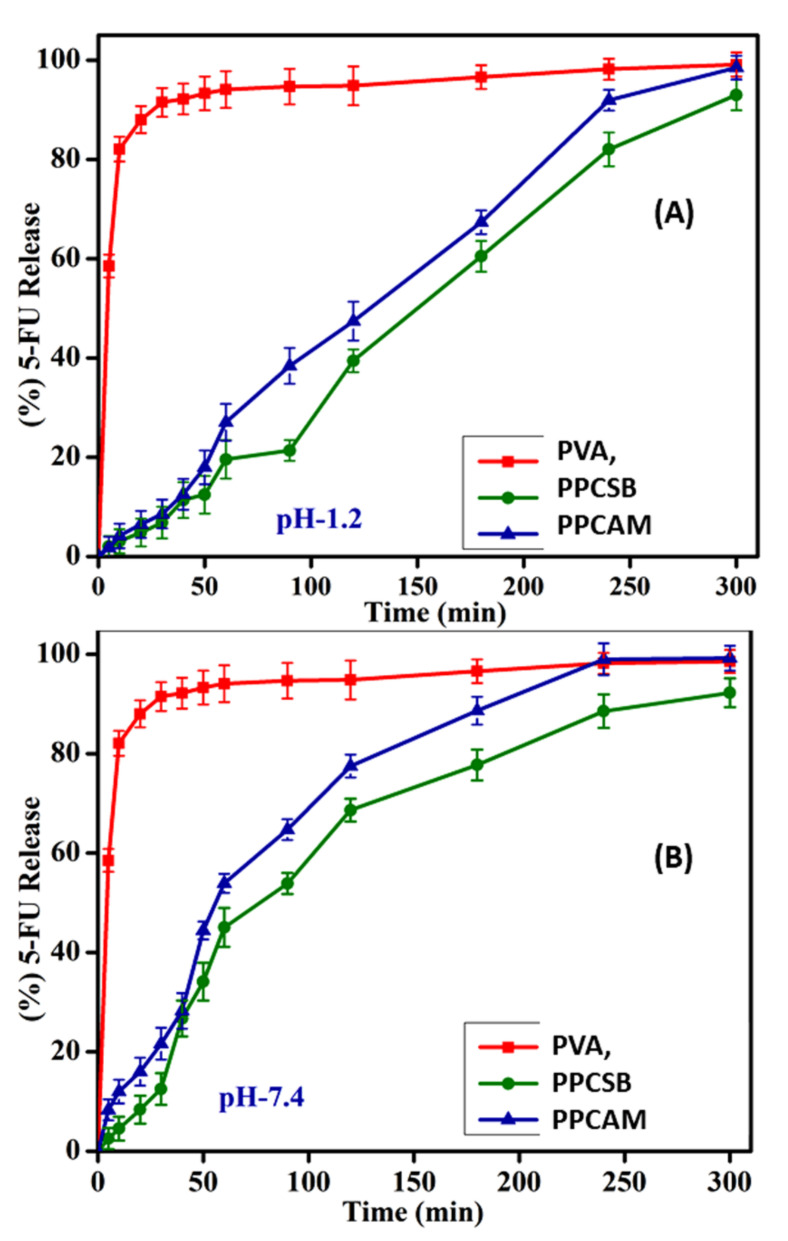
In vitro drug release studies of the PVA, PPCAM, and PPCSB membrane in pH-1.2 (**A**) and pH-7.4 (**B**). (Results denote mean ± S.D. (*n* = 3)).

**Figure 7 membranes-11-00792-f007:**
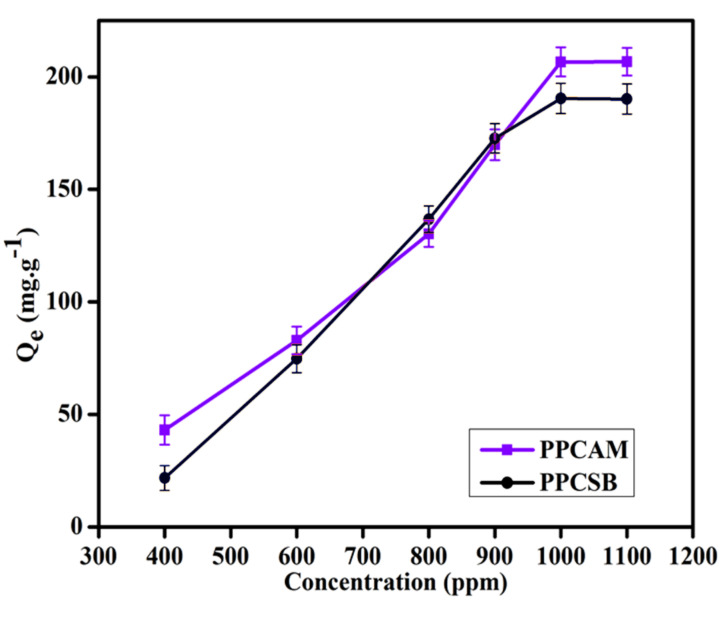
Copper ion adsorption by PPCAM and PPCSB PEMs at pH-5.5 (Results denote mean ± S.D. (*n* = 3)).

**Figure 8 membranes-11-00792-f008:**
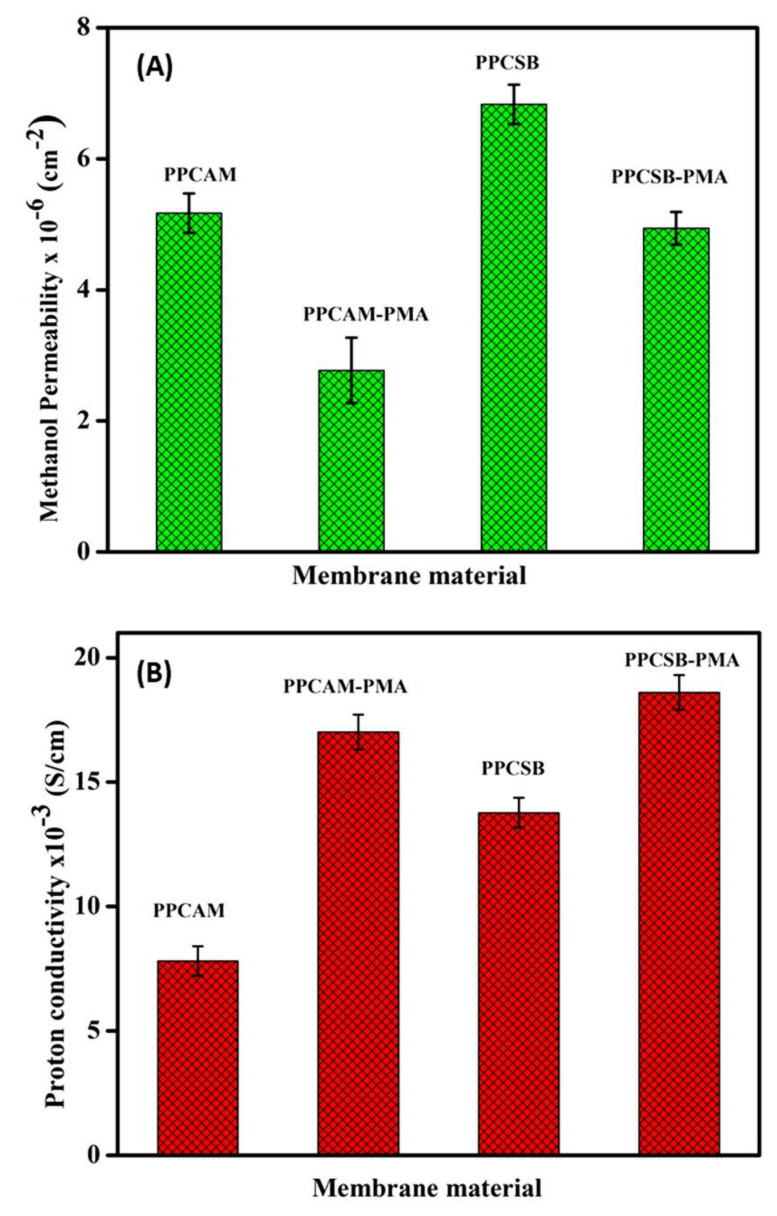
Methanol permeability capacities (**A**) and proton conductivity (**B**) of PPCAM and PPCSB PEMs.

**Table 1 membranes-11-00792-t001:** Composition, %S_e_, IEC, proton conductivity, and methanol permeability of various PEMs. (Results denote mean ± S.D. (*n* = 3)).

PEM Code	%ESR	IEC	Proton Conductivity (σ)∗10^−3^ (S/cm)	Methanol Permeability 10^−5^ (cm^−2^/s)
PPCAM	140	0.50	7.81	5.17
PPCAM-PMA	82	0.59	17.01	2.77
PPCSB	350	0.41	13.77	6.83
PPCSB-PMA	72	0.67	18.6	5.94

**Table 2 membranes-11-00792-t002:** Comparison of present work with pectin-based adsorbents (previous research).

Polymer	Metalion	Metal Uptake (mg/g)	Reference
Pectin-poly(Am-co-AGA) hydrogels	Cu^2+^	203.7	[21]
Pectin-iron oxide composite	Cu^2+^	48.99	[53]
Pectin based film	Cu^2+^	29.2	[54]
PPCSB	Cu^2+^	190.1	Present work
PPCAM	Cu^2+^	206.7	Present work

## Data Availability

Not applicable.

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
