# Peer review of "Fabrication of Polyelectrolyte Membranes of Pectin Graft-Copolymers with PVA and Their Composites with Phosphomolybdic Acid for Drug Delivery, Toxic Metal Ion Removal, and Fuel Cell Applications"

_membranes, 2021, doi:10.3390/membranes11100792_

Round 1

Reviewer 1 Report

This is an interesting and comprehensive investigation on Polyelectrolyte Membranes of Pectin Graft-co- 2 polymers with PVA and their Composites with Phosphomolyb- 3 dic Acid for Drug Delivery, Toxic Metal Ion Removal and Fuel  4 Cell Applications.

This Reviewer has just a few minor remarks:

  1. The English needs a thorough review, several sentences cannot be understood.
  2. The literature review should be expanded towards several types of membranes. I recommend adding this work: Castor oil and commercial thermoplastic polyurethane membranes modified with polyaniline: a comparative study
  3. Scheme 1 should be Figure 1. In addition, it should have a higher resolution.
  4. Equations 10,11 are not useful and should be deleted
  5. Figures 5,6,7: I suggest using more standard colours
  6. Please enhance both discussion of results and conclusions - both are too short.

Author Response

Reviewer 1

Comments and Suggestions for Authors

This is an interesting and comprehensive investigation on Polyelectrolyte Membranes of Pectin Graft copolymers with PVA and their Composites with Phosphomolybdic Acid for Drug Delivery, Toxic Metal Ion Removal and Fuel   Cell Applications.

This Reviewer has just a few minor remarks:

Authors are highly thankful to you and reviewers for valuable suggestions and comments to improve the quality of our manuscript.

  1. The English needs a thorough review, several sentences cannot be understood.

Answer: According to reviewer’s valuable suggestion, now authors are revised the manuscript by doing language correction with native English speaker Dr. Jayanth Kashyap, HOD, Senior Assistant Professor, Department of English, Yogi Vemana University, Kadapa.

2. The literature review should be expanded towards several types of membranes. I recommend adding this work: Castor oil and commercial thermoplastic polyurethane membranes modified with polyaniline: a comparative study

Answer: According

3. Scheme 1 should be Figure 1. In addition, it should have a higher resolution.

Answer: As per the reviewer comments now ‘Scheme 1’ is changed as ‘Figure 1’ with high resolution; accordingly other figures numbers also changed.

4. Equations 10,11 are not useful and should be deleted.

Answer: Equations 10 and 11 are essential in calculating the fuel cell properties i.e., ion-exchange capacity (IEC) and Proton conductivity (σ), respectively. Hence, Authors would like to retain the equations.

5. Figures 5,6,7: I suggest using more standard colours

Answer: According the reviewer suggestion, now authors are redrawn the Figures 5,6, 7 and replaced in the manuscript.

6. Please enhance both discussion of results and conclusions - both are too short.

Answer: According reviewers suggestion, now authors provided the necessary discussions.

Reviewer 2 Report

In the present study, the authors have fabricated polyelectrolyte membranes (PEMs) from pectin-graft-copolymers (PC-g-AMPS and PC-g-SVBS) and polyvinyl alcohol and their composites with phosphomolybdic acid and test their applications in drug delivery, toxic metal ion removal, and fuel cell applications. Although the prepared PEMs demonstrate promising results in the aforementioned applications, the goal and novelty of the paper is not very clear to me. It seems that authors tested the prepared PEMs for a bunch of applications. I think it would be helpful if the authors spend some time on rewriting the introduction and results section to clearly emphasize the goals of the paper. I, therefore, not recommend the publication of this manuscript in the current form. The authors need to significantly improve the manuscript for publication in this reputed journal.

Abstract:

  1. In the abstract, the authors summarize the results of their rstudy in the last 3-4 sentences. However, it is difficult to understand simply from the current abstract which values correspond to which type of PEMs. The authors should make this clear to the readers.
  2. It would be good if the authors clearly define the abbreviations at the first instance. For example, the authors introduce PEMs and PMA in the abstract, however, their definition is not provided until later in the introduction.

Introduction:

  1. In general, the introduction seems too long. However, it does not clearly emphasize the importance of the work presented here. I would recommend the authors to rewrite their introduction to clearly motivate the readers.
  2. Also, whenever references are made to the literature data, citations should be provided. A lot of citations are missing, for instance, in lines 107-110 and 119-120.

Experimental and Results:

  1. The biggest concern I have with the experimental and results section is that it is not clearly defined how many samples (n) were tested for each measurement to report the values presented in the current study. Do the presented values represent the mean? If yes, how many samples were tested? How were the error bars calculated (standard deviation, standard errors)? The authors should provide details in Figure and Table captions. Specific examples provided below:
    1. XRD patterns presented in Figure 2
    2. SEM images presented in Figure 3
    3. EDAX results in Figure 4 and in the text
    4. Values of %ESR, IEC, proton conductivity and methanol permeability in Table 1
    5. Drug release data presented in Figure 5
    6. Copper ion adsorption and metal ion uptake presented in Figure 6 and in Table 2
    7. Proton conductivity and methanol permeability values in Figure 7
  2. What is the molecular weight of the pectin polymer used in the present study? Currently, it just says medium. I am not sure what does that mean.
  3. The authors should clearly define the variables in equations (5)-(7). At this point, it is unclear what the different variables represent?
  4. In equation (1), there should be a factor of 100 on the right-hand side.
  5. In equation (3), it is not very clear what do the authors mean by theoretical loading. I would recommend the authors to clearly define this.
  6. In Figure 1, it may be better if the authors separate the results for PPCAM and PPCSB in two different panels to make it easier for the reader to follow.
  7. The scale bars are missing on the SEM images shown in Figure 3. The authors should clearly define the scale bars.
  8. I would recommend the authors to calculate the % crystallinity from the XRD patterns to better estimate how addition of PMA affects the crystallinity of PEMs prepared from PPCAM and PPCSB.
  9. In Figure 4, the EDAX data is missing for just PPCAM and PPCSB without any adsorption of 5-FU and Cu2+ metal ions.
  10. In lines 345-359, the authors mention that “The 5-FU release profiles indicated that drug release from PPCAM and PPCSB membranes are slightly higher in the pH 7.4 region when compared to the pH 1.2 region.” However, if I look at the graphs presented in Figure 5, it seems the other way around that the 5-FU release is faster at pH 1.2 in comparison to pH 7.4. The authors should double check the information presented, otherwise its confusing.
  11. The authors should define GA in line 151.

Author Response

Reviewer 2

Comments and Suggestions for Authors

In the present study, the authors have fabricated polyelectrolyte membranes (PEMs) from pectin-graft-copolymers (PC-g-AMPS and PC-g-SVBS) and polyvinyl alcohol and their composites with phosphomolybdic acid and test their applications in drug delivery, toxic metal ion removal, and fuel cell applications. Although the prepared PEMs demonstrate promising results in the aforementioned applications, the goal and novelty of the paper is not very clear to me. It seems that authors tested the prepared PEMs for a bunch of applications. I think it would be helpful if the authors spend some time on rewriting the introduction and results section to clearly emphasize the goals of the paper. I, therefore, not recommend the publication of this manuscript in the current form. The authors need to significantly improve the manuscript for publication in this reputed journal.

Authors are highly thankful to reviewer for their valuable suggestions and comments to improve the quality of the manuscript.

Abstract:

  1. In the abstract, the authors summarize the results of their study in the last 3-4 sentences. However, it is difficult to understand simply from the current abstract which values correspond to which type of PEMs. The authors should make this clear to the readers.

Answer: According to reviewers now added the sentences to the abstract about the type of PEMs

  1. It would be good if the authors clearly define the abbreviations at the first instance. For example, the authors introduce PEMs and PMA in the abstract, however, their definition is not provided until later in the introduction.

Answer:  Authors, now provided the abbreviations in the abstract.

Introduction:

  1. In general, the introduction seems too long. However, it does not clearly emphasize the importance of the work presented here. I would recommend the authors to rewrite their introduction to clearly motivate the readers.

Answer: According reviewers’ suggestion, now authors are emphasized the importance of the work as well as the reorganized the introduction part.

  1. Also, whenever references are made to the literature data, citations should be provided. A lot of citations are missing, for instance, in lines 107-110 and 119-120.

Answer: Now authors carefully cited the literature.

Experimental and Results:

  1. The biggest concern I have with the experimental and results section is that it is not clearly defined how many samples (n) were tested for each measurement to report the values presented in the current study. Do the presented values represent the mean? If yes, how many samples were tested? How were the error bars calculated (standard deviation, standard errors)? The authors should provide details in Figure and Table captions. Specific examples provided below:

    1. XRD patterns presented in Figure 2
    2. SEM images presented in Figure 3
    3. EDAX results in Figure 4 and in the text
    4. Values of %ESR, IEC, proton conductivity and methanol permeability in Table 1
    5. Drug release data presented in Figure 5
    6. Copper ion adsorption and metal ion uptake presented in Figure 6 and in Table 2
    7. Proton conductivity and methanol permeability values in Figure 7

Answer: Data presented in the manuscript is mean ±S.D. (n=3). The same is added at the appropriate figures and tables.

  1. What is the molecular weight of the pectin polymer used in the present study? Currently, it just says medium. I am not sure what does that mean.

Answer: Molecular weight of the pectin polymer is ‘30000-100000’ the same is incorporated in the materials part of the manuscript.

  1. The authors should clearly define the variables in equations (5)-(7). At this point, it is unclear what the different variables represent?

Answer: According reviewers’ suggestion, now clearly defined the variables in equations (5)-(7) as given below and the same is added in the manuscript.

Mt is the amount of 5FU release at time t, KH is the Huguchi rate constant, Q is the amount of the 5FU release at time t, Q0, is the amount of 5FU release time at t, K1 is the rate constant of first order and, K0 are the rate constant of zero order, KC is the 5FU release constant of Hixson-Crowell cube.

  1. In equation (1), there should be a factor of 100 on the right-hand side.

Answer: According reviewers’ suggestion, ‘100’ is added to the equation.

  1. In equation (3), it is not very clear what do the authors mean by theoretical loading. I would recommend the authors to clearly define this.

Answer: Equation is corrected for better understanding (‘theoretical loading’ is corrected as ‘Theoretical loading of 5-FU’

  1. In Figure 1, it may be better if the authors separate the results for PPCAM and PPCSB in two different panels to make it easier for the reader to follow.

Answer: According reviewers suggestion, Figure 1 ( in the revised manuscript Figure 2) is modified into two different panels. 

  1. The scale bars are missing on the SEM images shown in Figure 3. The authors should clearly define the scale bars.

Answer: According reviewers’ suggestion, now scale bars are Cleary defined in the revised figure.

  1. I would recommend the authors to calculate the % crystallinity from the XRD patterns to better estimate how addition of PMA affects the crystallinity of PEMs prepared from PPCAM and PPCSB.

Answer: Authors sincerely agree with the reviewers comment, but, due to non-availability of the XRD software, authors could not able to calculate the crystallinity at this moment, but we adopt this for our future studies.

  1. In Figure 4, the EDAX data is missing for just PPCAM and PPCSB without any adsorption of 5-FU and Cu2+ metal ions.

Answer: According to reviewers’ suggestion, now highlighted the Cu and 5-FU in the figure. Cu2+ adsorbed PPCAM (A), and Cu2+ adsorbed PPCSB (B), 5-FU encapsulated PPCAM (C), and 5-FU encapsulated PPCSB (D).

  1. In lines 345-359, the authors mention that “The 5-FU release profiles indicated that drug release from PPCAM and PPCSB membranes are slightly higher in the pH 7.4 region when compared to the pH 1.2 region.” However, if I look at the graphs presented in Figure 5, it seems the other way around that the 5-FU release is faster at pH 1.2 in comparison to pH 7.4. The authors should double check the information presented, otherwise its confusing.

Answer: Authors sincerely agree with the reviewers comment, authors regret to inform you that legends (pH 1.2 & pH 7.4) are wrongly assigned, now figure is corrected.

  1. The authors should define GA in line 151.

Answer: Now ‘GA’ is expanded as ‘glutaraldehyde’ and given in the manuscript.

Round 2

Reviewer 2 Report

The authors have addressed my comments, so the manuscript can be accepted in the present form.